

# Dark Matter phenomenology in 2HDMS

**Gudrid Moortgat-Pick[1,2]\*, Juhi Dutta[1], Cheng Li[2],
Merle Schreiber[1,2], Sheikh F. Tabira[1] and Julia Ziegler[1]**

**1** II. Inst. of Theo. Phys., University of Hamburg,
Luruper Chaussee 149, 22761 Hamburg, Germany
**2** Deutsches Elektronen-Synchrotron DESY,
Notkestr. 85, 22607 Hamburg, Germany

\* gudrid.moortgat-pick@desy.de

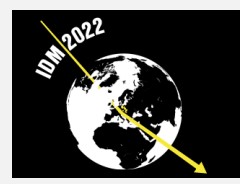

*14th International Conference on Identification of Dark Matter
Vienna, Austria, 18-22 July 2022*

## Abstract

**The constituents of dark matter are still an unresolved puzzle. Several Beyond Standard Model (BSM) Physics offer suitable candidates. In this study here we consider the Two Higgs Doublet model augmented with a complex scalar singlet (2HDMS) and focus on the dark matter phenomenology of 2HDMS with the complex scalar singlet as the dark matter candidate. The parameter space allowed from existing experimental constraints from dark matter, flavour physics and collider searches has been studied. The discovery potential for such a 2HDMS at HL-LHC and at future $e^+e^-$ colliders has been worked out.**

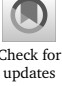

## 1 Introduction

Dark Matter (DM) remains an unsolved puzzle at the interface between particle physics and cosmology, only 4-5% of the Universe are composed by 'known' matter components, but about 25% is built of dark matter. Since the Standard Model (SM) does not accommodate a suitable DM candidate, several Beyond Standard Model (BSM) extensions have been proposed to accommodate DM candidates ranging from scalar, fermion to vector candidates and with mass scales from below eV up to TeV particles. We concentrate in this contribution on thermal weakly interacting massive particles (WIMP) that is expected in the mass range of GeV up to TeV, accessible at future collider experiments at the LHC and a high-energy $e^+e^-$ linear collider (ILC, CLIC).

Among popular BSM candidates are models with an extended Higgs sector such as the Two Higgs Doublet model (2HDM) [1], providing a dark matter candidate within the Inert Doublet model [1]. Alternate models are such multi-Higgs models but extended via real or complex singlet scalars serving as dark matter candidates. Such extensions involving real scalar singlets have been extensively studied [2–4] while complex scalar extensions to the 2HDM have also been recently studied in the context of modified Higgs sectors [5]. Such models have also the potential to explain the matter-antimatter asymmetry and to accommodate both inflation as well as gravitational waves phenomenology [6,7]. The parameter space of such extensions of the SM [8] gets strong constraints from direct searches for DM as well as from precision measurements of the 125 GeV SM-like Higgs boson and in particular from limits of both its visible as well as invisible branching ratios [9].

## 2 Extended two Higgs doublet model

### 2.1 Symmetries

We consider the CP-conserving softly broken Type II Two Higgs Doublet model augmented with a complex scalar singlet (2HDMS) [5] consistent with flavour changing neutral currents (FCNCs) at tree-level. It allows for the presence of the mixing term between the two Higgs doublets, $\Phi_1$ and $\Phi_2$, i.e., $m_{12}^2$, while the explicit $Z_2$ breaking terms are absent. The complex scalar singlet $S$ is stabilised by a $Z_2'$ symmetry such that $S$ is odd under $Z_2'$ while the SM fields are even under the new $Z_2'$ symmetry. The fields $\Phi_1$ and $S$ are even under $Z_2$ while $\Phi_2$ is odd under $Z_2$.

We consider the case where $Z_2'$ remains unbroken both explicitly and dynamically, i.e. the scalar singlet doesn't obtain a vacuum expectation value. Therefore, the scalar potential $V$ with a softly broken $Z_2$- and a conserved $Z_2'$ symmetry is $V = V_{2HDM} + V_S$, where, the softly broken $Z_2$-symmetric 2HDM potential is:

$$V_{2HDM} = m_{11}^2\Phi_1^\dagger\Phi_1 + m_{22}^2\Phi_2^\dagger\Phi_2 - (m_{12}^2\Phi_1^\dagger\Phi_2 + h.c) + \frac{\lambda_1}{2}(\Phi_1^\dagger\Phi_1)^2 + \frac{\lambda_2}{2}(\Phi_2^\dagger\Phi_2)^2 \tag{1}$$

$$+\lambda_3(\Phi_1^\dagger\Phi_1)(\Phi_2^\dagger\Phi_2) + \lambda_4(\Phi_1^\dagger\Phi_2)(\Phi_2^\dagger\Phi_1) + [\frac{\lambda_5}{2}(\Phi_1^\dagger\Phi_2)^2 + h.c], \tag{2}$$

and the $Z_2'$-symmetric singlet potential, $V_S$, is

$$V_S = m_S^2 S^* S + (\frac{m_S^{2'}}{2} S^2 + h.c) + (\frac{\lambda_1''}{24} S^4 + h.c) + (\frac{\lambda_2''}{6}(S^2 S^* S) + h.c) + \frac{\lambda_3''}{4}(S^* S)^2 \qquad (3)$$

$$+ S^* S[\lambda_1' \Phi_1^\dagger \Phi_1 + \lambda_2' \Phi_2^\dagger \Phi_2] + [S^2(\lambda_4' \Phi_1^\dagger \Phi_1 + \lambda_5' \Phi_2^\dagger \Phi_2) + h.c.]. \qquad (4)$$

The doublet fields have the components $\Phi_1 = (h_1^+, \frac{1}{\sqrt{2}}(v_1 + h_1 + ia_1))^T$, $\Phi_2 = (h_2^+, \frac{1}{\sqrt{2}}(v_2 + h_2 + ia_2))^T$, $S = \frac{1}{\sqrt{2}}(h_s + ia_s)$ and $\tan\beta = \frac{v_2}{v_1}$ is the ratio of the up-type and down-type Higgs doublet *vevs* $v_{1,2}$ (with $v(= v_1^2 + v_2^2) \simeq 246$ GeV. Under the assumption that the complex singlet scalar does not develop a *vev* —for this study imposed—, the Higgs sector, after EWSB, remains the same as in 2HDM, i.e, consisting of two CP-even neutral scalar Higgs particles $h$, $H$, a pseudoscalar Higgs $A$ and a pair of charged Higgs particles $H^\pm$ [1]. All Higgs-dark-matter portal couplings are explicitly given in [10].

## 2.2 Theoretical and phenomenological constraints

The Sylvester's criterion and copositivity [11, 12] has been applied to guarantee bounded-ness from below for the Higgs potential, leading to constraints on all coupling parameters $\lambda_i$, $i = 1,\ldots,5$, $\lambda_j'$, $j = 1,\ldots,5$, $\lambda_k''$, $k = 1,\ldots,3$. The mass of the lightest CP-even Higgs particle $m_h = 125$ GeV has been chosen to be in concordance with the measured Higgs state via HiggsSignals [13] and collider constraints from LEP and LHC have been applied for the heavy Higgs states via HiggsBounds [14]. The branching ratio $BR(h \to \chi\chi) < 0,11 (< 0.19)$, fulfilling the limits from ATLAS (CMS). Electroweak precision constraints on STU parameters have been taken into account as well as constraints from flavour physics $BR(b \to s\gamma)$, $BR(B_s \to \mu^+\mu^-)$, using SPheno [15]. $\Delta(g_\mu - 2)$. Concerning the dark matter particle, the bounds on the relic density from PLANCK measurements, $\Omega h^2 = 0.119$ [16], as well as constraints from direct detection (XENON-1T [17] ) and indirect detection (FERMI-LAT [18]) experiments have been applied using micrOMEGAs [19].

# 3 Results

## 3.1 Benchmark points

This model has been implemented using SARAH [20] code implemented into SPheno for the spectrum generation. In order to calculate collider observables the code chain Madgraph [21] -Pythia [22] -Delphes [23] -Madanalysis [24] has been used.

As can be seen from Figs.1a) and b) the mass of the dark matter particle $\chi$ as well as the coupling $\lambda_2'$ get strongest constraints from the direct detection search from XENON-1T: in the shown example where the heavier Higgs particles are about 725 GeV, the mass $m_\chi \sim 338$ GeV. Scanning the available parameter space allowed to specify different benchmark areas, see Table I. BP1 and BP3 are very similar, however, they differ significantly in the couplings $\lambda_1'$, $\lambda_2'$ and $\lambda_3$ leading to different collider phenomenology.

## 3.2 Collider phenomenology

In this section, we discuss the possible signals of this model at HL-LHC and future $e^+e^-$ colliders. As already mentioned, the invisible decay of the heavy Higgs into the dark matter candidate is a source of missing energy at colliders. Therefore, the direct production of heavy Higgs bosons and consequent decay of the Higgs to $\chi$ along with visible SM particles can give rise to distinct signatures for this scenario as opposed to the 2HDM like scenario. We investigate these possibilities and their prospects in the context of $\sqrt{s} = 14$ TeV LHC at the targeted

integrated luminosity of 3-4 ab$^{-1}$ and of future $e^+e^-$ colliders (ILC,CLIC) up to $\sqrt{s} = 3$ TeV and integrated luminosities of 5 ab$^{-1}$.

### 3.2.1 Prospects at LHC

The main processes contributing to neutral Higgs production at the LHC are gluon fusion (mediated by the top quark loop), vector boson fusion (VBF), associated Higgs production ($Vh_i$), $b\bar{b}h_i$, $t\bar{t}h_i$ [1]. For the charged Higgs pair, the possible production channels are $H^+H^-$ and $W^\pm H^\mp$ [1]. At the LHC Run 3 at $\sqrt{s} = 14$ TeV, all possible Higgs production processes (including SM and BSM Higgses) are summarised in Table II.

In the presence of the heavy Higgs $H$ decaying into two dark matter candidates, one gets invisible momentum in the final state and one can look into the following final states:

a)  $1j$ (ISR)+missing $E_T$ [25],

b)  $2j$+ missing $E_T$ [26] .

We estimate the significance for the mono-jet and VBF channels using the cuts from an existing cut-and-count analyses performed in [4] for $\sqrt{s} = 14$ TeV LHC, further details see [10]. For a) we achieve a cut efficiency for the signal in BP3 of about $\sim 18\%$ and we obtain a $0.111\sigma$ excess at 3ab$^{-1}$ using gluon fusion production channel (at leading order (LO)). For b) we get a signal efficiency of 4.5% for BP3 and a signal significance of about $\sim 0.2\,\sigma$ at 3 ab$^{-1}$. Therefore, we observe that due to the small invisible branching ratio and heavy Higgs masses $\sim 820$ GeV (and hence small production cross section) in BP3, the final states will be inaccessible at the upcoming HL-LHC run.

### 3.2.2 Prospects at a high-energy $e^-e^+$ Linear Collider (ILC, CLIC)

The cleaner environment and lower background along the beam line compared to hadron colliders make the electron-positron linear colliders an attractive choice for precision studies of new physics.

The International Linear Collider (ILC) [27] , is a proposed $e^+e^-$ linear collider design with simultaneously polarized $e^\pm$ beams and several stages of center-of-mass energies, i.e. at the SM-like Higgs threshold ($\sqrt{s} = 250$ GeV), at the top threshold ($\sqrt{s} = 350$ GeV) and at about $\sqrt{s} = 500$ GeV up to $\sqrt{s} = 1$ TeV with a maximum target integrated luminosity of $\mathcal{L} = 500$ fb$^{-1}$. The othe proposed high-energy $e^+e^-$ linear collider design is CLIC [28,29] with an energy upgrade up to $\sqrt{s} = 1.5, 3$ TeV and at least a polarized $e^-$beam. An overview of the physics potential at future high-energy linear colliders is given in [30]. ILC (CLIC ) gain advantage over the LHC in the possibility of exploiting the polarisation of the beams crucial both at the high energy stages but also already at the first stage of $\sqrt{s} = 250$ GeV [31, 32]. Although the invisible decay in BP3 is $H \to \chi\bar{\chi} \simeq 4.8\%$ and only a low production cross section times branching ratio is predicted, we observe that the $2b+$ missing $E_T$ channel is observable with a $= 3.99\sigma$ significance at an integrated luminosity of $\mathcal{L} = 5$ ab$^{-1}$.

# 4  Conclusions and Outlook

We have studied dark matter phenomenology in a Two Higgs Doublet model with a complex scalar singlet where the scalar singlet doesn't obtain a vacuum expectation value. Benchmark scenarios consistent with all current experimental and cosmological constraints have been worked out. Particular stringent bounds on the available parameter range for the couplings are set by bounds from the direct detection. Due to very small rates of the heavy Higgs production,

Table 1: Relevant parameters of the benchmark points used for the study [10].

| Parameters | BP1 | BP2 | BP3 |
|---|---|---|---|
| $\lambda_1$ | 0.23 | 0.1 | 0.23 |
| $\lambda_2$ | 0.25 | 0.26 | 0.26 |
| $\lambda_3$ | 0.39 | 0.10 | 0.2 |
| $\lambda_4$ | -0.17 | -0.10 | -0.14 |
| $\lambda_5$ | 0.001 | 0.10 | 0.10 |
| $m_{12}^2 (\text{GeV}^2)$ | $-1.0\times10^5$ | $-1.0\times10^5$ | $-1.0\times10^5$ |
| $\lambda_1''$ | 0.1 | 0.1 | 0.1 |
| $\lambda_3''$ | 0.1 | 0.1 | 0.1 |
| $\lambda_1'$ | 0.042 | 0.04 | 2.0 |
| $\lambda_2'$ | 0.042 | 0.001 | 0.01 |
| $\lambda_4'$ | 0.1 | 0.1 | 0.1 |
| $\lambda_5'$ | 0.1 | 0.1 | 0.1 |
| $\tan\beta$ | 4.9 | 6.5 | 6.5 |
| $m_h$ (GeV) | 125.09 | 125.09 | 125.09 |
| $m_H$ (GeV) | 724.4 | 816.4 | 821.7 |
| $m_A$ (GeV) | 724.4 | 812.6 | 817.9 |
| $m_{H^\pm}$ (GeV) | 816.3 | 816.3 | 822.2 |
| $m_\chi$ (GeV) | 338.0 | 76.7 | 323.6 |
| $\Omega h^2$ | 0.058 | 0.119 | 0.05 |
| $\sigma_p^{SI} \times 10^{10}$ (pb) | 0.76 | 0.052 | 2.9 |
| $\sigma_n^{SI} \times 10^{10}$ (pb) | 0.78 | 0.054 | 3.1 |

Table 2: The leading order (LO) cross section (in fb), further details see [10].

| Processes | Cross section (in fb) at $\sqrt{s} = 14$ TeV | | |
|---|---|---|---|
| | BP1 | BP2 | BP3 |
| $h$ (ggF) | $29.3\times10^3$ | $29.3\times10^3$ | $29.3\times10^3$ |
| $H$ | 22.61 | 5.238 | 6.632 |
| $A$ | 35 | 8.58 | 10.8 |
| $hjj$ (VBF) | $1.296\times10^3$ | $1.265\times10^3$ | $1.25\times10^3$ |
| $Hjj$ | 1.843 | 1.845 | 0.56 |
| $Ajj$ | 2.885 | 2.88 | 40.91 |
| $Wh$ | $1.148\times10^3$ | 1.133 | 1.134 pb |
| $WH$ | $1.195\times10^{-3}$ | 1.11e-03 | $1.199\times10^{-3}$ |
| $WA$ | $4.3\times10^{-4}$ | 5.892e-04 | $5.734\times10^{-4}$ |
| $Zh$ | 880.8 | 677.2 | 697.9 |
| $ZH$ | 0.93 | 0.2783 | 0.3408 |
| $ZA$ | 3.999 | 1.413 | 1.689 |
| $bbh$ | 2534 | 2541 | 2541 pb |
| $bbH$ | 21.52 | 17.92 | 17.92 fb |
| $bbA$ | 23.39 | 18.9 | 19.04fb |
| $t\bar{t}h$ | 478.3 | 477.1 | 477.9 |
| $t\bar{t}H$ | 0.1988 | 0.06571 | 0.7891 |
| $t\bar{t}A$ | 0.2552 | 0.08036 | 0.09826 |
| $H^+H^-$ | 0.06603 | 0.03033 | 0.03416 |
| $W^\pm H^\mp$ | 102.4 | 3.453 | 4.145 |
| $\chi\bar{\chi} + 1j$ | 0.006356 | 0.0681 | 0.8819 |

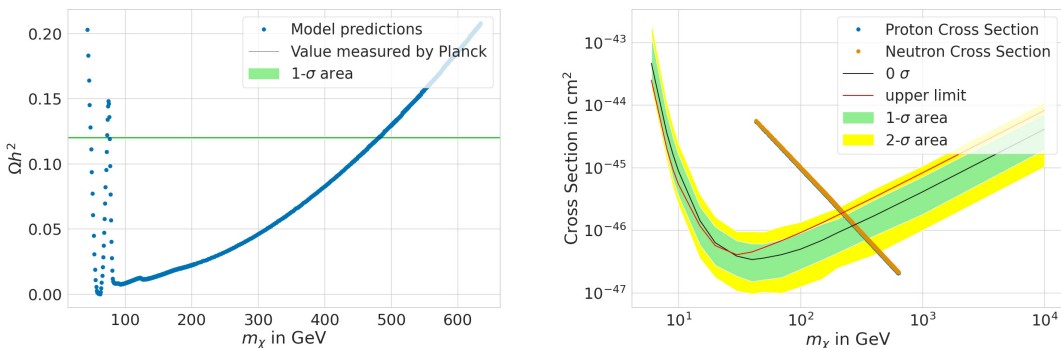

Figure 1: Relic density and direct detection cross-section predicted by the model depending on the DM mass $m_\chi$ [10]. The parameter $m_S^2$ has been varied in the range from 100-400000 GeV$^2$ and fixed $\tan\beta = 5$. The other parameters are chosen as in benchmark scenario BP1, see Table I.

the dark matter candidates will probably not be detectable via monojet or di-jet studies even at the HL-LHC. However, at a high-energy linear collider with polarized beams and precise initial energy, such a dark matter scenario is expected to be detectable. In addition, the option of direct dark matter pair production plus an ISR-photon is still under studies and offers another promising phenomenology. Further studies on exploring the mixing angles in the Higgs sector to shed light on the dark matter behaviour is still ongoing as well.

## Acknowledgments

**Funding information** JD and GMP acknowledge support by the Deutsche Forschungsgemeinschaft (DFG, German Research Foundation) under Germany's Excellence Strategy EXC 2121 "Quantum Universe"- 390833306.

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
