# Peer review of "Dark Matter Phenomenology in 2HDMS"

_SciPost Physics Proceedings, doi:SciPost Phys. Proc. 12, 072 (2023)_

## Round 1 · Referee Report · Anonymous · 2022-10-21

Strengths

1. The manuscript studies the dark matter phenomenology in the Higgs double model with an additional singlet with collider and direct/indirect detection bounds.

Weaknesses

Given the large number of free parameters, it is difficult to obtain conclusive results, or to provide more insights, for dark matter search.

Report

The study is solid, and provides enough constraints for this model , meeting the criteria of SciPost Physics Proceedings. So it can be published after several minor changes are made.

Requested changes

1. The parameters adopted for Fig.1 seem missing. The author should add the information here, otherwise, the reader can hardly get any information from the figure. A comment on Fermi-Lat bound would be nice, as the author claims that it is included.

2. Notations should be used consistently, such as "\emph{vev}" and "vev", also "missing ET" . BTW, all values of $\lambda$ should be real numbers (from Table I), why they have imaginary expressions just above section 2.2?

3. I strongly suggest the author improve the text, especially the last paragraph in Introduction.

---

## Round 2 · Referee Report · Anonymous (Referee 1) · 2023-1-10

Report

The manuscript has been improved, and can now be accepted in SciPost Physics Proceedings.

---

## Round 2 · Author Response

Dear Editors, we are very thankful to the referee for the careful proof reading of our manuscript. All issues have been addressed. In addition, minor language changes have been made and several references (in particular to the used codes etc. ) have been added.

---

## Editorial Decision

published